# New-Generation Liquid Crystal Materials for Application in Infrared Region

**DOI:** 10.3390/ma14102616

**Published:** 2021-05-17

**Authors:** Piotr Harmata, Jakub Herman

**Affiliations:** Faculty of Advanced Technologies and Chemistry, Military University of Technology, 2 gen. S. Kaliskiego St., 00-908 Warsaw, Poland; jakub.herman@wat.edu.pl

**Keywords:** nematic, ester, fluorinated liquid crystal, infrared transparent material, cladding layer, fused silica waveguides

## Abstract

This study presents 13 new organic compounds with self-assembling behavior, which can be divided into two groups. The first synthesized group includes compounds based on 4′-(trifluoromethoxy)-[1,1′-biphenyl]-4-yl-4-(trifluoromethoxy) benzoate core, and the second includes compounds based on 4-((4-(trifluoromethoxy)phenyl)ethynyl)phenyl-4-(trifluoromethoxy) benzoate core. They differ in the number and location of the fluorine atom in the lateral position. Mesomorphic properties, phase transition enthalpies, refractive indices, birefringence, and MWIR (mid-wavelength infrared) spectral properties of the compounds were investigated, and the results were compared with currently used materials. The influence of the length of the core as well as type and position of substituents in the molecular core was analyzed. The lack of aliphatic protons in the molecular structure generated unique infrared properties.

## 1. Introduction

The first thought that comes to mind at the mention of liquid crystals is the display industry [1,2,3,4], which has given them the greatest fame and recognition. It is not surprising as many items and devices of everyday use, such as monitors, TV sets, telephones, and smartwatches use these materials in the element designed for displaying information. Nevertheless, the application possibilities of liquid crystals (LC) go far beyond this area.

To better illustrate the breadth of this application, one should look at the spectrum of electromagnetic radiation. LCs are used in devices operating in visible light, infrared (IR) [5], millimeter wave [6,7,8,9,10], terahertz [11,12,13,14], and gigahertz [15,16,17,18].

In recent years, there has been an increased interest in the infrared range, in particular the MWIR (mid-wavelength infrared) range, where there are, among others, strategic communication bands [19,20] and photonic sensors for military applications [21,22,23,24,25,26]. It is worth noting that liquid crystals have already been used in the MWIR range in Lyot filters [27], photonic crystal fibers [28,29], light shutters [30,31,32], adaptive optics [33], laser beam steering [34,35,36], and optical valves [37,38,39]. However, their use has been limited to areas where LCs do not absorb radiation, i.e., windows. To increase the applicability of LCs in the MWIR range, the approach to molecular design has to be changed.

The vast majority of rod-like LCs synthesized so far are compounds from the so-called classical approach of molecular design. It assumes that in order for a compound to exhibit liquid crystal properties, it must consist of a rigid core and appropriate substituents in the terminal and lateral positions. The most commonly used cores include derivatives of oligophenyls (e.g., terphenyl [40,41] and quaterphenyl [42]) and tolanes (e.g., phenyltolane [43,44,45,46] and bistolane [47]), which may also have other linking groups, e.g., ester and ether [48]. To obtain a suitable shape anisotropy, aliphatic chain(s) and/or strong dipole moment generating groups, such as cyano (CN) [49] and isothiocyanate (NCS) [50,51], are used as terminal substituents. Laterally unsubstituted molecules often fail to meet the required criteria for LCs, such as low melting point, relevant mesomorphic morphology (generally only the nematic phase is desirable), smectic suppression, appropriate clearing temperature, sign of dielectric anisotropy, and sufficiently high optical anisotropy. Therefore, a fluorine [52,53,54] or chlorine atom is most often introduced into the rigid core in the lateral position [55]. In some cases, it is necessary to use larger bulky substituents, such as short alkyl chains in the form of methyl CH_3_ or ethyl C_2_H_5_ groups, to achieve the required properties. Molecules designed in this way can be used in all the aforementioned electromagnetic radiation ranges, excluding infrared, in particular the MWIR range. For these wavelengths, these materials show strong absorption bands [56]. The first band (λ ~ 3.0–3.5 µm) is very wide and of strong intensity, which comes from the stretching vibrations of the C–H bonds in the aromatic ring and the aliphatic chain. Another band is located at λ ~ 4.5 µm and is related to the stretching vibrations of C≡N and C≡C bonds. Moreover, the isothiocyanato (NCS) group has a strong and broad absorption band in the range of λ 4.5–5.2 µm. The presence of these parasitic absorption bands has led to very limited use of liquid crystals in the MWIR range.

As part of our previous work [57,58,59], we developed a new molecular design approach that allowed us to obtain a next-generation self-assembling material primarily for use in the infrared range, and particularly in the MWIR range [60,61,62,63]. This approach involved shifting parasitic absorption bands toward longer wavelengths. Therefore, in the terminal positions of rod-like molecules, the use of short perfluorinated chains and/or a fluorine or chlorine atom was proposed. In the lateral positions, the use of a fluorine atom, a chlorine atom, or a trifluoromethoxy group was proposed. This approach allowed us to obtain compounds characterized by high transparency in the MWIR range. It is worth emphasizing that it is not possible to completely eliminate absorption bands in this area while maintaining the mesomorphic properties of the compound. Moreover, previous studies have shown that with such short perfluorinated chains, the core is mainly responsible for the presence of liquid crystalline phases. Therefore, in this study, we decided to investigate its derivatives and try to broaden knowledge of the structure–properties relationship in this particular area of LCs that is still new.

High content of fluorine atoms in the molecule may have additional benefits. Such materials have been found to be used as a potential overlayer or cladding layer on fused silica waveguides [64,65,66]. Thus, it is possible to minimize dissipation losses. In order to achieve this goal, the liquid crystal must have one or both major refractive indices lower than that of the waveguide layer. Although such LC materials are known to exist [67,68], these compounds do not have conjugated (π-delocalized) systems and therefore have very low birefringence values (Δn ≈ 0.05). This study proposes a compound with a completely different structure that is rich in π-electrons, provides much higher birefringence, and meets the criteria for setting future waveguiding devices.

## 2. Experimental

### 2.1. Materials

1-Bromo-4-(trifluoromethoxy) benzene, 4-bromo-2-fluorophenol, 2-dicyclohexylphosphino-2′,6′-dimethoxybiphenyl (SPhos), 4-bromo-3-fluorophenol, 1-bromo-2-fluoro-4-(trifluoromethoxy) benzene, and oxone were purchased from Fluorochem (Hadfield, UK) and used as received. Magnesium for Grignard reactions (turnings), oxalyl chloride, 1,8-diazabicyclo(5.4.0)undec-7-ene, and 2-methylbut-3-yn-2-ol were purchased from Acros Organics, (Geel, Belgium) and used as received. Toluene, acetone, hydrochloric acid, sodium nitrite, potassium iodide, anhydrous potassium carbonate, potassium phosphate trihydrate, and copper(I) iodide were purchased from Avantor Performance Materials Poland S.A (Gliwice, Poland) and used as received. 4-Aminophenol, palladium(II) acetate, bis(triphenylphosphine)palladium(II) chloride, and sodium hydride (60% dispersion in mineral oil) were purchased from Sigma-Aldrich Sp. z.o.o (Poznan, Poland) and used as received. Pyridine was purchased from Lach-ner (Neratovice, Czech Republic) and used as received. Triethylamine was purchased from Chempur (Piekary Slaskie, Poland) and used as received. THF was distilled from sodium under nitrogen atmosphere prior to use. 2-(4-(Trifluoromethoxy)phenyl)-1,3,2-dioxaborinane, 2-(3-fluoro-4 (trifluoromethoxy)phenyl)-1,3,2-dioxaborinane, and 2-(3,5-difluoro-4-(trifluoromethoxy)phenyl)-1,3,2-dioxaborinane were synthesized in our laboratory; all the details of the synthesis and purification process are described in [58]. 1-Ethynyl-4-(trifluoromethoxy) benzene, 4-ethynyl-2-fluoro-1-(trifluoromethoxy) benzene, and 5-ethynyl-1,3-difluoro-2-(trifluoromethoxy) benzene were synthesized in our laboratory; all the details of the synthesis and purification process are described in [57].

### 2.2. Synthesis

In this work, 13 previously unpublished rod-like compounds were synthesized and characterized. The compounds were divided into two main groups. The first group consisted of compounds based on the 4′-(trifluoromethoxy)-[1,1′-biphenyl]-4-yl-4-(trifluoromethoxy) benzoate core, and the second group consisted of compounds based on 4-((4-(trifluoromethoxy) phenyl)ethynyl)phenyl-4-(trifluoromethoxy) benzoate (see Figure 1). Additionally compound 4-(trifluoromethoxy)phenyl-4-(trifluoromethoxy) benzoate was synthesized. Details of the synthetic procedures are given in the Appendix A.

The synthetic routes are shown in Figure 2, Figure 3, Figure 4, Figure 5, Figure 6 and Figure 7. The differences between the compounds resulted from the different location of the lateral substituents for a given core.

Two different synthetic methodologies were chosen to synthesize the 4′-(trifluoromethoxy)-[1,1′-biphenyl]-4-yl-4-(trifluoromethoxy) benzoate-based compounds. The first methodology (Figure 2 and Figure 5) involved the synthesis of the appropriate derivative 4′-(trifluoromethoxy)-[1,1′-biphenyl]-4-ol and 4-(trifluoromethoxy) benzoyl chloride. The last step was the esterification reaction, which led to the formation of the final product. This approach is used very often but requires the separate synthesis of the appropriate phenol each time lateral substitution is changed. The second methodology (Figure 3 and Figure 4) involved preparing 4-halophenyl-4-(trifluoromethoxy) benzoate and applying it in the Suzuki–Miyaura (SM) coupling reaction to obtain the final product. This approach seems more versatile as the resulting aryl halide can also be used in other coupling reactions. The second advantage is the high commercial availability of boronic acids. The main disadvantage of this approach is the possibility of an ester hydrolysis reaction in the SM coupling conditions.

For the synthesis of compounds based on the group 2 core 4-((4-(trifluoromethoxy)phenyl)ethynyl)phenyl-4-(trifluoromethoxy) benzoate (Figure 5) and the compound 4-(trifluoromethoxy) phenyl 4-(trifluoromethoxy) benzoate (Figure 6), only one synthetic variant was carried out because these methodologies seem to be the most optimal and allow use of intermediates synthesized in the earlier stages.

#### 2.2.1. Synthesis of 4′-(Trifluoromethoxy)-[1,1′-biphenyl]-4-yl-4-(trifluoromethoxy) Benzoate Derivatives

The synthesis of compound **8a** (Figure 2, variant A) started with the preparation of 4-(trifluoromethoxy) benzoic acid **2**. For this, commercially available 1-bromo-4-(trifluoromethoxy) benzene was used, which was converted into Grignard reagent followed by the addition of solid CO_2_ at −78 °C. The obtained acid **2** was transformed into acid chloride **3**. The obtained product was poured into a volumetric flask, and the concentration was determined (C = 0.87 mol/dm^3^). Product **3** was used for all syntheses in this work. The second key intermediate was biphenol derivative **7a**. Its synthesis began with the use of commercially available 4-aminophenol, which was converted into a diazonium salt and then into 4-iodophenol **5**. The next step was the SM coupling of the previously synthesized boronic ester **6a** and 4-iodophenol **5**. In this way, two key intermediates were synthesized, namely **3** and **7a**. The last step to obtain the final compound was the esterification reaction.

The synthesis of compounds **8b** and **8c** was performed using a different path (Figure 3, variant B). An esterification reaction was carried out using the previously synthesized intermediates **3** and **5**. The 4-iodophenyl-4-(trifluoromethoxy) benzoate **9** obtained in this way was subjected to an SM coupling reaction with the previously prepared boroorganic esters **6b** and **6c**. This course of action allowed two final compounds to be obtained.

Structures **13a**–**13d** was initially planned to be synthesized using variant B. However, unexpected difficulties were encountered during the synthesis. Using the commercially available 4-bromo-3-fluorophenol **10a** and the earlier obtained 4-(trifluoromethoxy) benzoyl chloride **3**, intermediate **11** was synthesized in the first stage (Figure 4). However, the last step of the reaction turned out to be problematic. Under the SM coupling conditions, ester **11** was subjected to the hydrolysis process of ester linkage. Only compound **13b** was obtained with this method. In the remaining cases, it was not possible to extract the final compounds, despite attempts to change the reaction conditions. Therefore, compounds **13a**, **13c**, and **13d** were obtained using variant A. To obtain compounds **13a**, **13c**, and **13d** (Figure 5), the corresponding 4′-(trifluoromethoxy)-[1,1′-biphenyl]-4-ol derivatives had to be obtained. Therefore, for this purpose, the SM coupling reaction of bromophenols **10a** and **10b** with previously synthesized organoboron esters **6a** and **10c** was performed. The last step to obtain the final compound was the esterification reaction.

#### 2.2.2. Synthesis of 4-((4-(Trifluoromethoxy)phenyl)ethynyl)phenyl-4-(trifluoromethoxy) Benzoate Derivatives

The synthesis of compounds **18a**–**18d** was performed as follows (Figure 6). In the first step of this synthesis, a monoprotected acetylene derivative was introduced into the aromatic ring. In the case of synthesized compounds, the source of the carbon–carbon triple bond was 2-methyl-3-butyn-2-ol **15**, which is the cheapest commercially available monoprotected acetylene derivative. The second step was deprotection of the obtained derivative by the hydrolysis reaction using a catalytic amount of a base, in this case sodium hydride. The most commonly used solvent is toluene. Its boiling temperature is higher than the temperature of the removable protection (acetone), which is distilled off during the reaction, so sodium hydride can be added in catalytic amounts. This way, four key intermediates, namely **17a**–**17d**, were obtained. The last step to obtain the final compound was Sonogashira coupling.

#### 2.2.3. Synthesis of 4-(Trifluoromethoxy)phenyl-4-(trifluoromethoxy) Benzoate

The last compound synthesized in this work was compound **20**. It was prepared (Figure 7) by converting boronic ester **6b** to phenol using oxidation reaction with commercially available oxone. Then, the final reaction was the ester linkage creation.

### 2.3. Characterization and Measurements—Analytical Instrumentation

Synthesis progress and purity of synthesized compounds were determined using Shimadzu GCMS-QP2010S series (Shimadzu, Kyoto, Japan) gas chromatograph equipped with a quadrupole mass analyzer (MS(EI)), high-performance liquid chromatography (HPLC–PDA–MS (APCI-ESI dual source)) using Shimadzu LCMS 2010 EV (Shimadzu, Kyoto, Japan) equipped with a polychromatic UV–vis detector (Shimadzu, Kyoto, Japan), and thin-layer chromatography (silica gel on aluminum). IR spectra (region 2–6 µm) of the compounds as solutions in CCl_4_ (C = 0.05 mmol/cm^3^) in square 10 mm cuvettes were collected at 25 °C using Thermoscientific Nicolet iS10 (Thermo Fisher Scientific, Waltham, MA, USA). The phase transition temperatures and enthalpy data were determined by polarizing optical microscopy (POM) with an Olympus BX51 (Olympus, Shinjuku, Tokyo, Japan) equipped with a Linkam hot-stage THMS-600 (Linkam Scientific Instruments Ltd., Tadworth, UK) and differential scanning calorimeter SETARAM DSC 141 (KEP Technologies Group’s DNA, Montauban, France) during heating/cooling cycles (with rate of 2°/min). Refractive indices of neat materials and multicomponent nematic mixture were measured by the Metricon Model 2010/M Prism Coupler (Metricon Corporation, Pennington, NJ, USA) equipped with 443, 636, and 1550 nm lasers. Samples of liquid crystals were placed on glass plates rubbed with SE-130 polyimide (Nissan Chemical^®^, Tokyo, Japan). Ordinary refractive index n_o_ and extraordinary refractive index n_e_ were measured separately using different polarization of incident beams. Samples were measured at elevated temperatures (5 Kelvin steps), with the limit of the setup being 200 °C.

### 2.4. Mesomorphic Properties and Discussion

Temperatures and enthalpies of phase transitions of synthesized compounds are listed in Table 1.

Due to the location of the lateral substituents in the core, two position types were distinguished, namely inner core (red) and outer core (green) substituents. As mentioned earlier, two series of compounds were synthesized (see Figure 1), and these series had different features. The common feature for both series was the presence of the CF_3_OPhCOO group, in which no modifications were made. However, there were differences between these compounds that influenced the rest of the molecules. Group 1 included compounds in which the benzene rings were connected directly, and functionalization was performed only in the lateral position (acronyms **8a**–**13e**). The second group included compounds in which a carbon–carbon triple bond was introduced between benzene rings, and functionalization was also performed only in the lateral position (acronyms **18a**–**18d**). Moreover, an additional compound **20** was synthesized, which allowed us to find the correlation between the structure of the compound and its mesomorphic properties. This was one of the components of the multicomponent mixture, which helped to lower the clearing temperature and enhanced the miscibility between materials.

To better illustrate the correlations, the results are presented in the Figure 8, Figure 9, Figure 10, Figure 11 and Figure 12. Figure 8 shows the influence of the number and position of fluorine atoms in the lateral position on the mesomorphic properties for selected derivatives of group 1.

The unsubstituted compound **8a** showed a high melting temperature (above 140 °C). A smectic A phase occurred, and a very narrow (3.3 °C) nematic phase range was observed. Compound **8a** could be treated as a core and primary structure of group 1 compounds **8b**–**13e**, which differed in the number and location of fluorine atoms in the lateral position. The introduction of the fluorine atom **8b** in the outer core position caused a decrease in the melting temperature by 36.6 °C. Moreover, destabilization of the highly ordered smectic phase A was noticed, which resulted in the temperature range of the nematic phase increasing significantly by 24.9 °C. The subsequent introduction of a fluorine atom in the outer core position **8c** further lowered the melting point and narrowed the temperature range of the nematic phase.

Figure 9 shows the next observed correlations resulting from the number and location of fluorine atoms in the core.

For a detailed analysis of the properties, the internal separation of substituents were introduced at the inner core positions into A and B, marked in orange and red, respectively, in Figure 10. Compounds **13a**–**c**, in which the fluorine atom was directed toward the adjacent benzene ring, were collected in set B. Set A included compounds **13e** and **13d**, which had a lateral substituent directed toward the ester group. The introduction of one fluorine atom **13a** into the rigid core showed the stabilizing effect of the smectic phase (its temperature range did not change). However, the melting temperature dropped significantly by 58.0 °C and widened the range of the nematic phase by 12.7 °C. In the case of compound **13e**, where there was also only one fluorine atom (inner core position A), a significant extension of the temperature range of the smectic phase by 22.0 °C and destabilization of the nematic phase were observed. Moreover, the melting point practically did not change (compared to **13a**). Further introduction of the fluorine atom in the outer core position (**13b** and **13d**) caused a drastic change in the mesophase morphology. In the case of compound **13b**, the smectic phase disappeared completely, the range of the nematic phase was significantly widened by 26.0 °C, and the melting point was lowered. However, for compound **13e**, the observations were slightly different. First of all, no drop in the melting point was observed, the range of the smectic phase was significantly narrowed by 57.0 °C, and the nematic phase also appeared. Subsequent introductions of a fluorine atom in the outer core position (**13c**) increased the melting point and narrowed the range of the nematic phase. It is worth noting that the compounds from set A had a much stronger tendency to create smectic phases. One general conclusion that can be drawn from the figures above is that if the lateral substituent (fluorine atom) was in the inner core position (both A or B), the compounds tended to form smectic phases. The reverse was true if the substituent was in the outer core position, with increased nematogeneity being observed. A similar relationship was found for compounds from group 2, which had carbon–carbon triple bond introduced between benzene rings (acronyms **18a**–**18d**) (see Figure 11). Taking into account the stability of the nematic phase, the optimal number of fluorine atoms in a rigid core should not be greater than 2.

Figure 11 shows the influence of the number and position of fluorine atoms in the lateral position on the mesomorphic properties of tolane derivatives.

The unsubstituted compound **18a** showed a melting point above 110 °C, a temperature-wide smectic phase A (79.0 °C range), and a temperature-narrow nematic phase. Compound **18a** could be somehow treated as a source of compounds **18b**–**18d**, which differed in the number and location of fluorine atoms in the lateral position. The introduction of one fluorine atom into the core had a dramatic effect on the phase appearance. The fluorine atom in the outer core position (**18b**) strongly destabilized the presence of the smectic phase. Moreover, the temperature range of the nematic phase was significantly extended by 63.8 °C, and the melting point was lowered. On the other hand, when the fluorine atom was in the inner core position (**18d**), the compound exhibited only a smectic phase in a very wide range (129.1 °C) and even lower melting point. Further introduction of the fluorine atom in the outer core position (**18c**) caused the complete disappearance of the smectic phase, lowering the melting point and narrowing the range of the nematic phase by 13.5 °C.

Figure 12 shows the influence of the core structure on the mesomorphic properties.

As described in previous works [57,58,59] in the case of mesomorphs dedicated to infrared applications, where short perfluorinated chains are used, the presence of liquid crystalline phases is mainly related to the structure of the core itself. Functionalization in the lateral position helps to fine-tune expected parameters such as mesophase morphology, temperature range, and clearing and melting temperatures. Figure 12 confirms this observation. With the elongation of the rigid core, the temperature range of the liquid crystalline phase increased. Moreover, the above described correlations clearly show the crucial role of the number and position of the lateral substituents on the type of liquid crystalline phases that occur.

### 2.5. IR (2–6 μm) Absorption Properties

One of the goals of this research was the synthesis of new compounds that show reduced absorption in the infrared range, especially in the range of 2–6 μm (MWIR). Due to the fact that the obtained spectra were virtually identical for each of the compounds investigated here, separate IR spectra for each compound are not shown. Figure 13 shows the transmittance spectra for selected compounds from groups 1 and 2 (blue and green lines). Two previously reported compounds have also been added to the graphics to show the influence of the core structure on properties in the range of 2–6 µm (black and red lines). Moreover, the properties of the compounds investigated here were correlated with the most commonly used nematic 5CB (4-cyano-4′-pentylbiphenyl; show as orange line in the figure). The choice of 5CB was not accidental. It was chosen because its structure, i.e., a rigid core and the presence of an elastic aliphatic chain and a group generating a strong dipole moment, is common to the vast majority of liquid crystals that have been synthesized and used so far. This allowed visualization of the differences between the “older generation” materials and those dedicated to infrared. First, the absorption bands associated with infrared-transparent compounds were analyzed.

The absorption bands, shown in Figure 13, were closely related to their structure. In the case of compounds dedicated to infrared, the strongest absorption band at λ ~ 5.7μm (1754 cm^−1^) came from the basic vibrations of the C=O bond from the carbonyl group. Another narrow, moderately intense absorption band at λ ~ 4.5 μm (2128 cm^−1^) was associated with the presence of the C≡C bond. However, the intensity of these vibrations changed with the anharmonicity change of the oscillator, which was the R1-C≡C-R2 system, as detailed in [57]. The different intensities of this band for different structures containing the C≡C triple bond were also directly related to the presence of fluorine atoms in positions proximal to the OCF_3_ group [62]. There were also quite wide, low intensity peaks at around λ ~ 4.2 μm (2381 cm^−1^), which originated from the overtone C–F and C–O bonds. Basic vibrations for C–F bond (CF, CF_2_, and CF_3_) were in the range of 7.1–9.5 μm (1408–1052 cm^−1^), while basic vibrations for single C–O bond were in the range of 7.68–8.65 μm (1302–1156 cm^−1^). It had a small absorption peak at around λ ~ 3.3 μm (3030 cm^−1^) derived from the C–H bond vibrations in the aromatic rings. The last peak of very low intensity, which was placed at λ ~ 2.9 μm (3448 cm^−1^), originated from the overtone C=O.

Comparing the spectra of compounds dedicated to infrared and 5CB, it was found that the very wide and strong intensity peak at λ ~ 3.0–3.5 (3333–2857 cm^−1^) came mainly from the vibrations of C–H bonds in the aliphatic chain. The second peak, which was narrow and also of strong intensity, was related to the CN group. This indicated that the change in molecular design resulted in a new generation of infrared-dedicated materials. Moreover, the obtained ester derivatives, due to their structure, had additional absorption bands. However, they could still be treated as materials showing high transparency in the infrared range.

### 2.6. Optical Properties

Six investigated structures were selected for measurement of optical properties. The exact values of refractive indices n_o_ and n_e_ are given in the Appendix A. Figure 14 shows a graphical comparison of the measured values for six selected structures, namely **8b**, **8c**, **13b**, **13c**, **18b**, and **18c**.

With the optical data obtained for 636 nm for five tested compounds, namely **8b**, **8c**, **13b**, **18b**, and **18c** (the nematic shown by compound **13c** was too narrow), we estimated the macroscopic order parameter S using the Haller extrapolation, which proposed the following relation [69]:(1)$$S={\left(1-\frac{T}{T_{\mathrm{NI}}}\right)}^{\beta }$$
where T/T_NI_ is reduced temperature, and β is an adjustable parameter. Anisotropic properties such as birefringence ∆n and its temperature dependence in the nematic phase is given by the following relation: [70]
(2)$$∆n=∆n_{0}{\left(1-\frac{T}{T_{\mathrm{NI}}}\right)}^{\beta }$$
where ∆n_o_ is the value of the birefringence in perfect order at 0 K. The parameters ∆n_o_ and β were obtained by fitting the experimental data for ∆n to the following equation written in logarithmic form (3):(3)$$\log∆n=\log∆n_{0}+\;\beta \;\log\;(\frac{T_{\mathrm{NI}}-T\;}{T_{\mathrm{NI}}})$$

Then, the required order parameter S was obtained by substituting this value in (1). Variation of S values with reduced temperature (T/T_NI_) for five investigated structures is given in Figure 15.

Comparing the biphenyl benzoate structures (group 1), it was found that each successive fluorine atom in the rigid core of the molecule caused a decrease in the value of birefringence Δn for each wavelength. The same behavior was noticed among structures with triple C≡C bond (group 2). The presence of a triple bond in the core of the molecule caused an increase in the value of extraordinary refractive index, and thus birefringence, compared to the biphenyl benzoate structures. The values of order parameter for all samples were found to gradually decrease with increase in temperature. The tested compounds were characterized with standard values of parameter S (0.3 < S < 0.7) in nematic phases; however, compound **8b** differed from the others, with slightly higher values of S.

The additional advantage of the liquid crystals tested here is the very low values of ordinary refractive indices, which are lower than those found for fused silica [71,72,73,74].

However, measurements were made at elevated temperatures, well above room temperature. Encouraged by this fact, we composed a nematic multicomponent mixture from the examined structures that covered temperatures even below room temperature. The weight composition of the nematic multicomponent mixture at room temperature (T_Cr-N_ < 0 °C, T_N-Iso_ = 52.0–54.9 °C) is given in the Appendix A.

For the test mixture, measurements of the refractive indices were also made for three wavelengths (443, 636, and 1550 nm) in a wide temperature range. The refractive index data for the multicomponent mixture are given in the Appendix A.

With the values of the refractive indices n_e_ and n_o_ for any three wavelengths, we used the methods of mathematical model fitting to the experimental results to determine the exact values of the three coefficients of the Cauchy equation (Equation (4)) A_e,o_, B_e,o_, and C_e,o_ given in Table 2:(4)$$n_{e,o}={\;A}_{e,o}+\frac{B_{e,o}}{\lambda^{2}}+\frac{C_{e,o}}{\lambda^{4}}$$

With Cauchy coefficients, we further calculated the refractive index and birefringence dispersion in a large range of electromagnetic spectrum (see Figure 16). At room temperature (20 °C), the material was again found to have low ordinary refractive indices, with values below those reported for fused silica for some spectral range.

Such materials are desirable for use as an overlayer or cladding layer on fused silica waveguides [64,65,66]. So far, most of the known liquid crystal materials showing a similar effect were built from hydrocarbons without π-delocalized bonded groups [67,68]. Accordingly, the birefringence values of such LCs were very low (Δn ≈ 0.05). In this work, the liquid crystal structures showed a completely different molecular approach. Cores of the molecules were made of only aromatic rings as well as triple C≡C bonds rich in π-electrons. This in turn translated into much higher values of birefringence.

## 3. Conclusions

As part of this study, 13 previously unpublished compounds were obtained, which were characterized by reduced infrared absorption, in particular in the range of 2–5.7 μm. Based on the physicochemical research performed, the following conclusions can be drawn:-Of the synthesized compounds, 12/13 had enantiotropic liquid crystalline phases.-Compounds with inner core lateral substitution tended to form temperature-stable smectic phases. The situation was different in the case of outer core, where nematic phases were observed.-From the viewpoint of the nematic phase, the optimal number of fluorine atoms in a rigid core should not be greater than 2.-The presence of absorption bands in the range of 2–6 µm is inevitable if a compound is to exhibit mesomorphic properties. Most often, these bands come from the vibrations of the C–H bonds from the rigid core or the C–O and C–F bonds related to the functionalization in the terminal and lateral positions.

## Figures and Tables

**Figure 1 materials-14-02616-f001:**
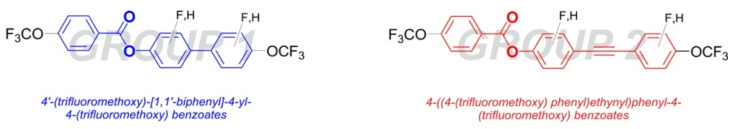
General structure of synthesized compounds.

**Figure 2 materials-14-02616-f002:**
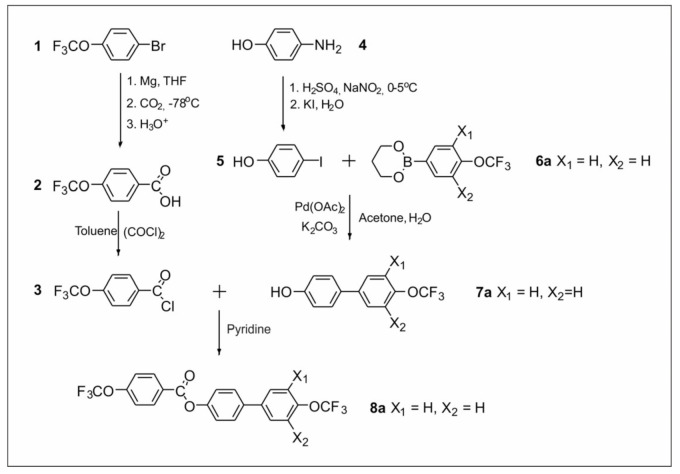
Synthesis of 4′-(trifluoromethoxy)-[1,1′-biphenyl]-4-yl 4-(trifluoromethoxy) benzoate **8a** (variant A).

**Figure 3 materials-14-02616-f003:**
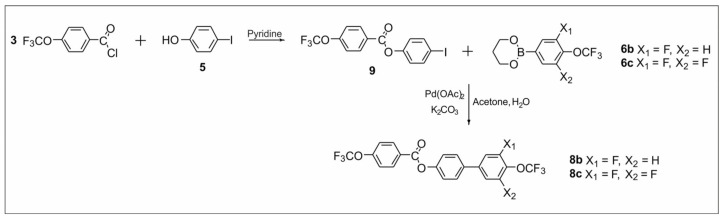
Synthesis of ester derivatives **8b** and **8c** (variant B).

**Figure 4 materials-14-02616-f004:**
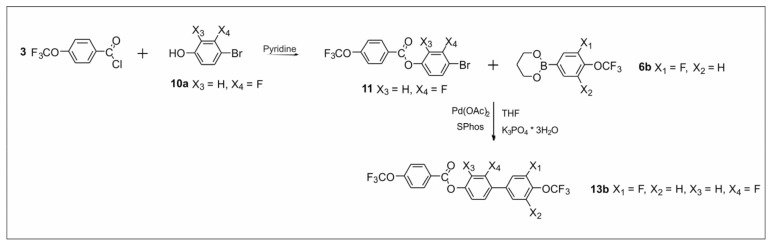
Synthesis of 2,3′-difluoro-4′-(trifluoromethoxy)-[1,1′-biphenyl]-4-yl 4-(trifluoromethoxy) benzoate **13b**.

**Figure 5 materials-14-02616-f005:**
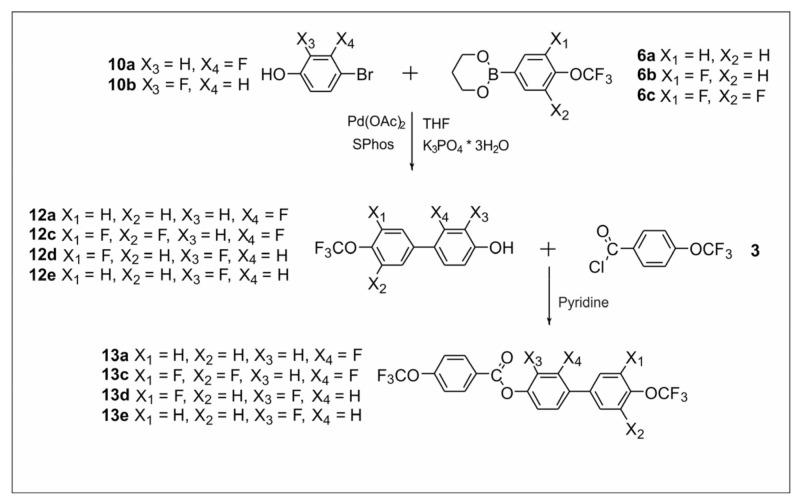
Synthesis of ester derivatives **13a**, **13c**, **13d**, and **13e**.

**Figure 6 materials-14-02616-f006:**
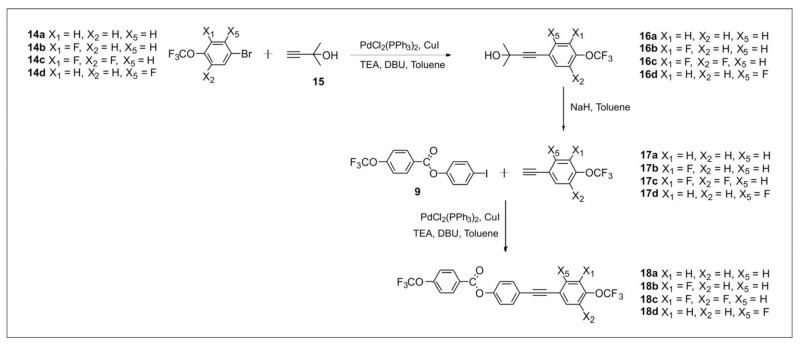
Synthesis of ester–tolane derivatives **18a**–**18d**.

**Figure 7 materials-14-02616-f007:**
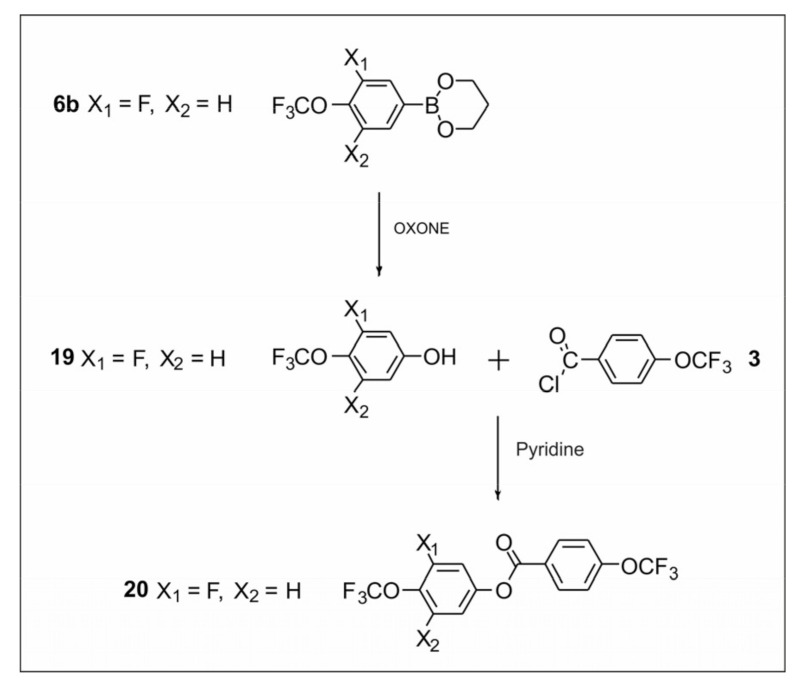
Synthesis of 3-fluoro-4-(trifluoromethoxy)phenyl 4-(trifluoromethoxy) benzoate **20**.

**Figure 8 materials-14-02616-f008:**
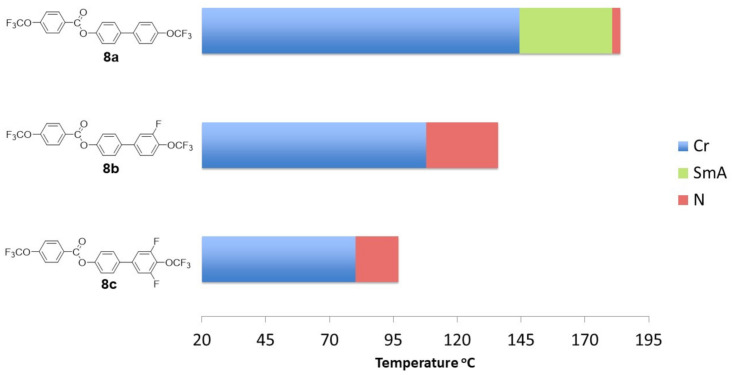
The influence of the number and position of fluorine atoms in the lateral position on the mesomorphic properties for selected group 1 derivatives (direct connection between benzene units).

**Figure 9 materials-14-02616-f009:**
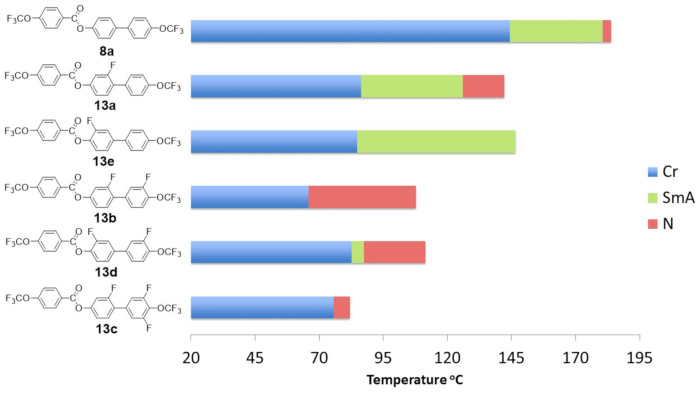
The influence of the number and position of fluorine atoms in the lateral position on the mesomorphic properties for selected group 1 derivatives (direct connection between benzene units).

**Figure 10 materials-14-02616-f010:**
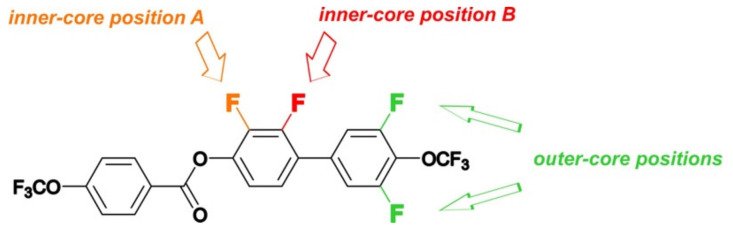
Positions of fluorine substituents in 4′-(trifluoromethoxy)-[1,1′-biphenyl]-4-yl-4-(trifluoromethoxy) benzoates (group 1 compounds): inner core position A (orange), inner core position B (red), and outer core position (green).

**Figure 11 materials-14-02616-f011:**
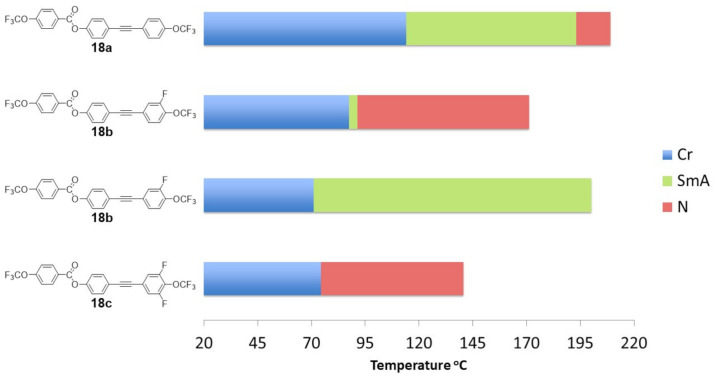
The influence of the number and position of fluorine atoms in the lateral position on the mesomorphic properties of tolane derivatives.

**Figure 12 materials-14-02616-f012:**
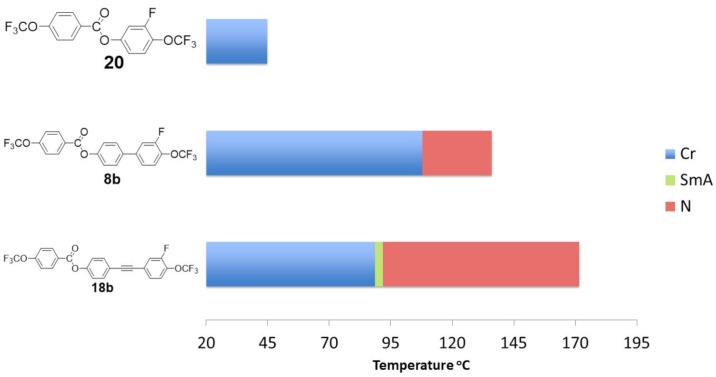
The influence of the core structure on the mesomorphic properties.

**Figure 13 materials-14-02616-f013:**
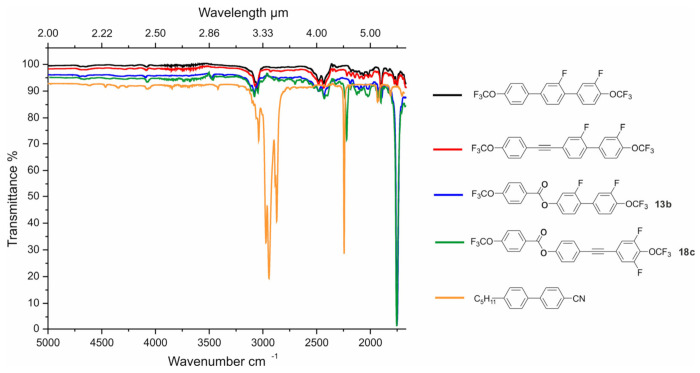
Spectral properties in the range of 2–6 μm for selected compounds: terphenyl (in black) [57], phenyltolane (in red) [58], **13b** (in blue), **18c** (in green), and 5CB (in orange).

**Figure 14 materials-14-02616-f014:**
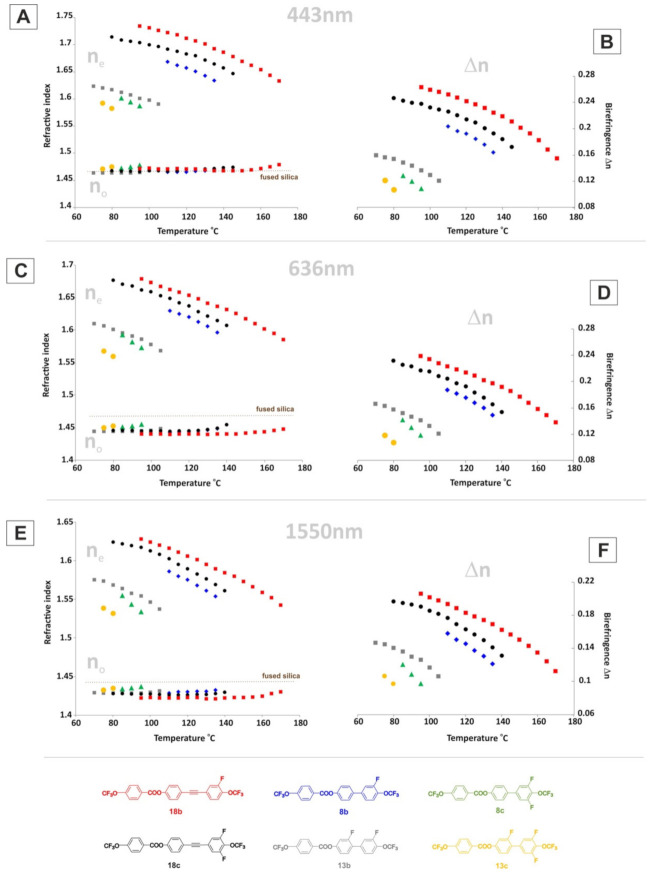
Temperature dependence of refractive indices n_o_ and n_e_ (**A**,**C**,**E**) and birefringence Δn (**B**,**D**,**F**) measured for **8b**, **8c**, **13b**, **13c**, **18b**, and **18c** at λ = 443, 636, and 1550 nm.

**Figure 15 materials-14-02616-f015:**
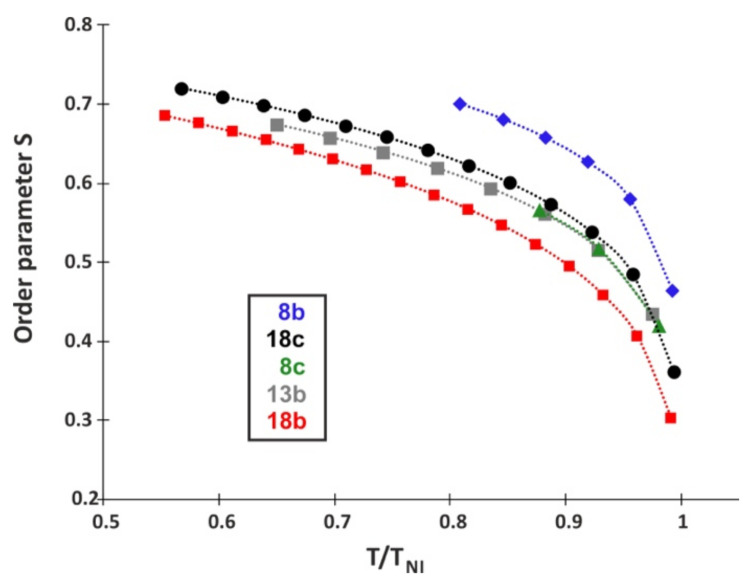
Temperature dependence of order parameter S for **8b**, **8c**, **13b**, **18b**, and **18c**.

**Figure 16 materials-14-02616-f016:**
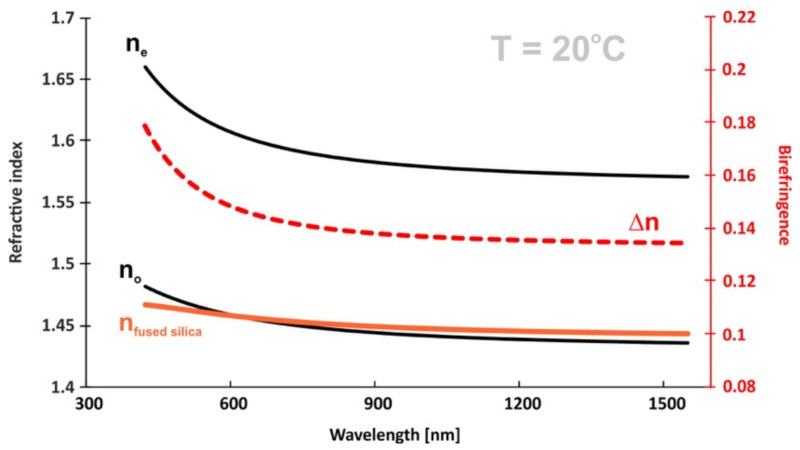
Dispersion of refractive indices (black) and birefringence (red) for multicomponent nematic mixture and refractive index of fused silica (brown) at 20 °C.

**Table 1 materials-14-02616-t001:** Temperatures and enthalpies of phase transitions for synthesized compounds.

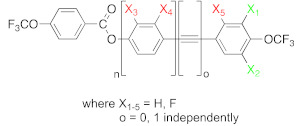														
**Compound Number**	**n**	**o**	**X_1_**	**X_2_**	**X_3_**	**X_4_**	**X_5_**	**Temperatures and Enthalpies of Phase Transitions**						
								**Cr**	**(°C)** **(kJ/mol)**	**SmA**	**(°C)** **(kJ/mol)**	**N**	**(°C)** **(kJ/mol)**	**Iso**
**8a**	1	0	H	H	H	H	H	*	144.4 22.66	*	180.6 2.88	*	183.9 1.33	*
**8b**	1	0	F	H	H	H	H	*	107.8 27.55	–		*	136.0 0.60	*
**8c**	1	0	F	F	H	H	H	*	80.2 24.91	–		*	96.9 0.42	*
**13a**	1	0	H	H	H	F	H	*	86.4 18.25	*	126.1 1.16	*	142.1 0.87	*
**13b**	1	0	F	H	H	F	H	*	65.8 17.19	–		*	107.8 0.51	*
**13c**	1	0	F	F	H	F	H	*	75.7 29.23	–		*	81.9 0.35	*
**13d**	1	0	F	H	F	H	H	*	82.7 24.67	*	87.4 0.17	*	111.4 0.85	*
**13e**	1	0	H	H	F	H	H	*	84.8 21.43	*	146.5 3.92	–		*
**18a**	1	1	H	H	H	H	H	*	114.1 28.14	*	193.1 1.23	*	209.5 0.92	*
**18b**	1	1	F	H	H	H	H	*	87.5 27.84	*	91.3 0.19	*	171.1 0.69	*
**18c**	1	1	F	F	H	H	H	*	74.3 24.13	–		*	140.6 0.54	*
**18d**	1	1	H	H	H	H	F	*	71.1 20.03	*	200.2 4.65	–		*
**20**	0	0	F	H	H	H	H	*	44.8 23.19	–		–		*

* means that a given phase is present; – means that a given phase does not exist.

**Table 2 materials-14-02616-t002:** Fitting parameters for the extended Cauchy model at 20 °C using the experimental data measured at 443, 636, and 1550 nm wavelengths for multicomponent nematic mixture.

Temperature [°C]	n_o_			n_e_		
	A_o_	B_o_	C_o_	A_e_	B_e_	C_e_
20	1.4323	1.0331 × 10^4^	−2.7761 × 10^8^	1.5654	1.3305 × 10^4^	6.4903 × 10^8^

## Data Availability

Data is contained within the article or Appendix A.

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
