# Peer review of "New-Generation Liquid Crystal Materials for Application in Infrared Region"

_materials, 2021, doi:10.3390/ma14102616_

Round 1

Reviewer 1 Report

The submitted paper reports on the design and characterization of liquid crystal materials tailored to mid-infrared wavelength applications. The authors synthesized 13 new mesogenic materials and studied their phase transitions and optical properties (MIWR spectra, temperature dependent refractive indices). The authors succeeded in finding the correlations between the chemical structure and MIWR absorption spectra. In addition, they managed to prepare a room-temperature liquid crystal mixture using the synthesized materials. The paper is well written with conclusions supported by the performed experiments. I support the publication of this paper. I think it will be of high interest to professionals working in this field.  There are minor comments the authors could consider:

Minor comments:

The authors could also indicate the clearing point corresponding to the nematic phase – isotropic phase transition. By looking at Figures 8-12, it is not clear whether the authors explored the isotropic phase or just reached the clearing point. At what temperature does the decomposition of the synthesized materials take place?

Figure 13: Indicate the thickness of the studied liquid crystal samples. In addition, add their temperature.

Figure 17 shows a room temperature liquid crystal mixture. It covers visible and near IR diapason (up to 1500 nm). How does it behave in MWIR (1500 nm – 5000 nm)?

Author Response

We would like to thank for the time and effort invested in the review of our manuscript, and for helpful comments and suggestions. We revised the manuscript with special attention to all received comments and we believe that the Reviewer’s suggestions have improved our manuscript. In the following, we have addressed all comments as can be seen in the enclosed list. The original Reviewer’s comments are shown in italics and changes and correction inside manuscript are marked in yellow. 

Review 1

The submitted paper reports on the design and characterization of liquid crystal materials tailored to mid-infrared wavelength applications. The authors synthesized 13 new mesogenic materials and studied their phase transitions and optical properties (MIWR spectra, temperature dependent refractive indices). The authors succeeded in finding the correlations between the chemical structure and MIWR absorption spectra. In addition, they managed to prepare a room-temperature liquid crystal mixture using the synthesized materials. The paper is well written with conclusions supported by the performed experiments. I support the publication of this paper. I think it will be of high interest to professionals working in this field.  There are minor comments the authors could consider:

Minor comments:

The authors could also indicate the clearing point corresponding to the nematic phase – isotropic phase transition. By looking at Figures 8-12, it is not clear whether the authors explored the isotropic phase or just reached the clearing point. At what temperature does the decomposition of the synthesized materials take place?

Figures 8-12 show the clearing temperatures, which correspond to the exact values presented in Table 1. Individually, each compound has not been tested for thermal stability, however, two of the obtained structures have stable liquid crystalline phases up to 200oC. No thermal decomposition was observed for them at this temperature. Therefore, it was assumed that also the remaining compounds show high thermal stability, even at temperatures close to 200°C.

/ Figure 13: Indicate the thickness of the studied liquid crystal samples. In addition, add their temperature.

IR measurements were made from solutions in CCl4 with a concentration (0.05mmol/cm3) in square 10mm cuvettes at temperatures (25oC). The information is attached in the manuscript, section 2.3.

Figure 17 shows a room temperature liquid crystal mixture. It covers visible and near IR diapason (up to 1500 nm). How does it behave in MWIR (1500 nm – 5000 nm)?

Our measurement setup performs the measurements up to 1550nm. However, the exact values of the Cauchy equation coefficients Ae,o ; Be,o and Ce,o, were added to the text manuscript, on the basis of which the behavior of the multicomponent material in MWIR can be estimated

Reviewer 2 Report

Manuscript "A new generation liquid crystal materials for applications in infrared region”, written by Piotr Harmata and Jakub Herman, was submitted for publication in Materials.

Authors are reporting 13 new liquid crystalline compounds with the aim to evaluate their mesomorphic and optical properties. I appreciate the synthetic work. I am sure that the authors present high quality work in chemical synthesis. As well, optical properties of new compounds are valuable.

Before being published, I recommend to do a revision of the presented manuscript. I have several remarks and suggestions:

In table 1, are authors sure that the enthalpies are in J/mol? I would expect kJ/mol. Please, add melting points, they are more important than the temperatures of crystallization (these ones are well defined only at selected conditions).

Line 425: Indexes “0” are not in the subscript positions.

Figure 13: I would prefer to add numbers of studied compounds and references for compounds studied previously.

The interpretation of absorption spectra with respect to vibrations of various linkages and/or groups is not ideal at all. If “C-triple bond-C” linkage is active at 4.5micrometeres, why is it not present for the second compound (red line) with the same bond? I am completely confused and I am not convinced that the interpretation what authors are presenting concerning IR properties in relevance to specific vibrations, that it is really true. The scale in reciprocal “cm” (for example in brackets after the value in micrometers) would be valuable as people working in spectroscopy are more familiar with this scale units. This part is the weakest point of the manuscript and it should be rewritten.

In Figure 14, I do not find necessary to present again the chemical formula below the graphs. Chemical formulas are in Figure 8,9 and 11. It is really not necessary to repeat chemical formulas in every Figure.

Figure 16: I am sure this Figure is redundant. Compounds obtained labels previously and formulas are not necessary to be repeated. The weight in % can be added into text or in a table to be put into Supplementary File. To be frank, I do not understand what type of multicomponent mixtures authors prepared and discussed.

In summary, the presented manuscript is original and present a good piece of synthetic work. I can recommend the manuscript for publication only after a revision.

Author Response

We would like to thank for the time and effort invested in the review of our manuscript, and for helpful comments and suggestions. We revised the manuscript with special attention to all received comments and we believe that the Reviewer’s suggestions have improved our manuscript. In the following, we have addressed all comments as can be seen in the enclosed list. The original Reviewer’s comments are shown in italics and changes and correction inside manuscript are marked in yellow. 

Review 2

Manuscript "A new generation liquid crystal materials for applications in infrared region”, written by Piotr Harmata and Jakub Herman, was submitted for publication in Materials.

Authors are reporting 13 new liquid crystalline compounds with the aim to evaluate their mesomorphic and optical properties. I appreciate the synthetic work. I am sure that the authors present high quality work in chemical synthesis. As well, optical properties of new compounds are valuable.

Before being published, I recommend to do a revision of the presented manuscript. I have several remarks and suggestions:

In table 1, are authors sure that the enthalpies are in J/mol? I would expect kJ/mol. Please, add melting points, they are more important than the temperatures of crystallization (these ones are well defined only at selected conditions).

The reviewer accurately noticed the error in the unit of the melting enthalpy. It should indeed be the kJ/mol unit, which has been corrected in Table 1. The table shows the melting points that are determined during the Cr-N (or Cr-SmA) transitions in the heating cycles. Table 1 does not include data on crystallization temperatures obtained from cooling cycles.

Line 425: Indexes “0” are not in the subscript positions.

Reviewer is right. Information was corrected in the manuscript.

Figure 13: I would prefer to add numbers of studied compounds and references for compounds studied previously.

Numbers of studied compounds and references are added to the Figure 13 and its caption.

The interpretation of absorption spectra with respect to vibrations of various linkages and/or groups is not ideal at all. If “C-triple bond-C” linkage is active at 4.5micrometeres, why is it not present for the second compound (red line) with the same bond? I am completely confused and I am not convinced that the interpretation what authors are presenting concerning IR properties in relevance to specific vibrations, that it is really true. The scale in reciprocal “cm” (for example in brackets after the value in micrometers) would be valuable as people working in spectroscopy are more familiar with this scale units. This part is the weakest point of the manuscript and it should be rewritten.

The exact interpretation of the transmittance intensity for the triple bond (CC) in the compared compounds is given in ref [58] and [62]. The different intensity of this band for different structures containing the CC triple bond is also directly related to the presence of fluorine atoms in positions proximal to the OCF3 group. The more fluorine atoms next to the OCF3 group, the greater the intensity of the observed band. The description of IR in the manuscript has been enriched to include these explanations with relevant literature references.

In Figure 14, I do not find necessary to present again the chemical formula below the graphs. Chemical formulas are in Figure 8,9 and 11. It is really not necessary to repeat chemical formulas in every Figure.

In order to better understand the information contained in the manuscript and better catch the structure-properties correlation, we intentionally included multiple repetitions of structures in graphs and drawings, which eliminates the need to search for the correct formula each time.

Figure 16: I am sure this Figure is redundant. Compounds obtained labels previously and formulas are not necessary to be repeated. The weight in % can be added into text or in a table to be put into Supplementary File. To be frank, I do not understand what type of multicomponent mixtures authors prepared and discussed.

According the reviewer suggestions Figure 16 is transferred to Suplementary Materials, and the text in manuscript contains only necessary information about multicomponent nematic blend.

In summary, the presented manuscript is original and present a good piece of synthetic work. I can recommend the manuscript for publication only after a revision

Reviewer 3 Report

The authors present the manuscript "A new generation liquid crystal materials for applications in infrared region". The manuscript is interesting, well written and well organized.

I have two minor comments:

1) Can the authors say something on the possible agreement between the Equations 2 and 3, which model the dependence on Δn upon the temperature, and the data reported in Figure 14 for different liquid crystals at different wavelengths?

2) The authors state that they "determined 424the exact values of the three coefficients of the Cauchy equation Ae,o, Be,o, and Ce,o" (line 424 and 425). It would be beneficial to report these values in the manuscript such that the reader can use these values for further studies.

Author Response

We would like to thank for the time and effort invested in the review of our manuscript, and for helpful comments and suggestions. We revised the manuscript with special attention to all received comments and we believe that the Reviewer’s suggestions have improved our manuscript. In the following, we have addressed all comments as can be seen in the enclosed list. The original Reviewer’s comments are shown in italics and changes and correction inside manuscript are marked in yellow. 

Review 3

The authors present the manuscript "A new generation liquid crystal materials for applications in infrared region". The manuscript is interesting, well written and well organized.

I have two minor comments:

  • Can the authors say something on the possible agreement between the Equations 2 and 3, which model the dependence on Δn upon the temperature, and the data reported in Figure 14 for different liquid crystals at different wavelengths?

We used birefringence data in the direct extrapolation method to estimate the order parameter. For the purpose of (S) the term log Δn has been plotted as a function of log (TNI-T/TNI) – equation 3. The plot becomes linear. β values were obtained by fitting the experimental data for Δn to the equation 2 written in the logarithmic form. β values became close to 0,17 through the linear regression.

2) The authors state that they "determined 424the exact values of the three coefficients of the Cauchy equation Ae,o, Be,o, and Ce,o" (line 424 and 425). It would be beneficial to report these values in the manuscript such that the reader can use these values for further studies.

We added to the manuscript the exact values of the Extended Cauchy model coefficients Ae,o ; Be,o and Ce,o – Table 2